

# Experimental assessment of diffusible iodine-based contrast-enhanced computed tomography (diceCT) protocols

Lucy F. Costello[1], Hazel L. Richards[2], Alistair R. Evans[2,3] and Justin W. Adams[1,2]

[1] Anatomy and Developmental Biology, Monash University, Clayton, Victoria, Australia
[2] Geosciences, Museums Victoria, Melbourne, Victoria, Australia
[3] School of Biological Sciences, Monash University, Clayton, Victoria, Australia

## ABSTRACT

Diffusible iodine-based contrast-enhanced computed tomography (diceCT) is an increasingly used digital complement, supplement, or alternative to traditional dissection-based anatomical research. The diceCT protocol, which has evolved and expanded over the past decade, employs passive diffusion of Lugol's iodine ($KI_3$) to increase soft tissue radiodensity and improve structure contrast in the CT or microCT imaging of specimens. The development and application of diceCT has focused largely on specimens under 1 kg, and the varying reporting of methods on studies of both small and large specimens has initiated, but not yet established, an effective diceCT protocol for larger specimens based on monitored experiments of several fundamental variables (*e.g.*, Lugol's iodine concentration, duration, and impacts of Lugol's iodine on tissues). In this study, we have experimentally assessed the efficacy of diceCT protocols for imaging whole-body specimens of the 1–4.5 kg Australian brushtail possum (*Trichosurus vulpecula*) using sequential CT imaging assessment across experimental conditions. We assessed the impact of varying Lugol's iodine concentration, the presence/absence of skin, solution volume and agitation on tissue radiodensity changes through weekly CT-based monitoring of tissue radiodensities over an 8-week experimental period. We have also quantified tissue volumetric changes across our experiment to assess the impact of diceCT applications on subsequent analyses of imaging datasets. Our results indicate that substantial changes in both soft-tissue radiodensity and soft-tissue volume occur within the first 28 days of Lugol's iodine treatment, followed by a slower rate of progressive soft-tissue radiodensity and volume changes across the experiment duration. Our results demonstrate the negligible benefit of skinning larger specimens to improve solution diffusion, and document significant soft-tissue volumetric changes with high concentration solutions (*e.g.*, 10%) and long-duration exposure (*e.g.*, beyond 5 weeks) that should guide individual diceCT protocol design and/or quantification and analysis for mammal specimens above 1 kg.

Corresponding author
Lucy F. Costello,
lucy.costello@monash.edu

## INTRODUCTION

Anatomical research has undergone significant transformation in recent decades through the progressive adoption of 'cyberanatomy' methods that unite medical and advanced imaging technologies with computational approaches to quantify organism structure and support integrated analysis of morphology (*e.g.*, *Plyusnin et al., 2008*; *Cox & Faulkes, 2014*; *Lautenschlager, Bright & Rayfield, 2014*; *Dickinson et al., 2019*; *Demuth et al., 2022*). One significant methodological advance, building from decades of progress with computed tomography (CT) in medical and non-medical applications, was the development of diffusible iodine-based contrast enhanced computed tomography (diceCT) (*Metscher, 2009a*, *2009b*; *Gignac et al., 2016*). This specimen treatment protocol uses the passive diffusion of Lugol's iodine ($KI_3$) to differentially increase the radiodensity of soft tissues that are generally poorly visualised with traditional CT imaging. This development, and subsequent use and experimentation with diceCT treatment protocols, has fostered a significant expansion in soft-tissue visualization, quantification and research across taxonomic groups (*Metscher, 2009a*, *2009b*; *Gignac & Kley, 2014*; *Li et al., 2015*; *Gignac et al., 2016*; *Bribiesca-Contreras & Sellers, 2017*; *Hedrick et al., 2018*; *Fahn-Lai, Biewener & Pierce, 2020*; *Lanzetti & Ekdale, 2021*; *Richards, Adams & Evans, 2023*).

The rapid adoption and expansion across taxonomic groups has understandably led to diverse diceCT approaches, but equally significant is variability in the reporting of how or why specific steps in specimen treatment are taken in each study. This variable reporting can either be taken as reflecting the results of unpublished or uncommented-on experimentation in refining diceCT applications to specific research questions; or equally simply reflecting usage of previously published protocols that may or may not be optimal for particular specimens or novel research questions. For example, early applications of diceCT were largely focused on whole or partial organisms under 1 kg in mass (*Cox & Jeffery, 2011*; *Degenhardt et al., 2010*; *Fernandez et al., 2014*; *Gignac Paul & Kley Nathan, 2018*; *Holliday et al., 2013*; *Jeffery et al., 2011*; *Lautenschlager, Bright & Rayfield, 2014*; *Nasrullah, Renfree & Evans, 2018*; *Stephenson et al., 2012*; *Tahara & Larsson, 2013*; *Wong et al., 2012*; *Wong, Spring & Henkelman, 2013*), with minimal experimental assessment with larger animals. Simultaneously, published diceCT studies (whether topic-or methods-based) have not fully addressed to what extent skinning specimens enhances Lugol's iodine diffusion, or the timing and extent of specimen shrinkage with varying diceCT treatment conditions (*Vickerton, Jarvis & Jeffery, 2013*; *Wong et al., 2012*; *Hedrick et al., 2018*). Furthermore, optimising diceCT protocols to minimise specimen impacts (such as tissue shrinkage) is particularly important for studies aiming to quantify soft-tissue structures, apply consistent protocols across multiple studies, and when study design includes eventual longer-term retention of specimens (*e.g.*, specimens from or intended to be reposited in museum collections).

Here we report a series of experimental diceCT protocols using whole-body adult specimens of the common brushtail possum (*Trichosurus vulpecula*; Class Marsupialia, Order Diprotodontia, Family Phalangeridae). With a typical adult body mass between 1–4.5 kg and crown-rump length of 30–50 cm (*Silva & Downing, 1995*; *Nowak, 1999*),

brushtail possums represent a larger mammal species than commonly published in the experimental development of diceCT protocols; yet they overlap in mass, overall size and body proportions with a range of mammals of common interest to anatomical researchers. We investigate the effects of skinning, Lugol's iodine stain concentrations, and staining duration on specimen radiodensity and tissue preservation through repeated, sequential CT imaging and analysis. We then discuss the results of our varying experimental conditions to provide recommendations on diceCT protocols and considerations when interpreting the outcomes of diceCT imaging datasets.

## MATERIALS AND METHODS

Two brushtail possums (*Trichosurus vulpecula*, MUTV1C and MUTV2C; see discussion of specimen acronyms below) were obtained as cadaveric tissues under the Victorian Department of Environment, Land, Water & Planning Wildlife Act of 1975 Research Permit number 10008717, with the remaining MUTV2 cadaveric specimens purchased commercially.

Our experimental approach consisted of two stages. Stage One tested protocols from the published literature to establish the duration of Lugol's iodine exposure required to effect radiodensity increases across an entire full body (>1 kg) specimen, and to assess any tissue shrinkage. The results gathered from Stage One informed the development of novel experimental protocols that were applied in Stage Two. In Stage Two, we varied and evaluated the impact of different diceCT specimen treatments (presence of skin, Lugol's iodine concentration, solution agitation, and solution volume) on qualitative and quantitative measures *via* CT imaging.

It is important to note that the specimens were eviscerated *via* an abdominopelvic incision to mimic typical museum preparation, and to minimize handling issues (*e.g.*, abdominal content herniation) given the repeated, sequential CT scanning undertaken in our experiment. While allowing for the free circulation of Lugol's iodine into the abdominopelvic cavity, this process reduced the overall specimen mass from ~2–3 kg to ~1.2 kg (see Table 1 for individual specimen body masses). We would note, however, that while our experimental specimens ultimately weighed ~1.2 kg, the muscles being evaluated in this study are from an animal with recorded liveweights that are generally 2–3 times greater.

The specimens have been given abbreviations based on their protocols (*e.g.*, MUTV#X), where the 'MUTV' prefix denotes 'Monash University *Trichosurus vulpecula*', the number (#) records the applicable experiment stage (1 or 2), and the letter suffix (X) correspond to the protocol. For example, MUTV1C is the specimen from experiment stage 1 and is the control. A correspondence table of experiment stages, protocols and specimen abbreviations is provided in Table 1.

### Experimental design

#### *Stage one*

Our first experimental specimen was a skinned and eviscerated (*via* a single midsagittal abdominopelvic incision) adult common brushtail possum that had been stored at −20 °C
**Table 1 The MUTV2 specimen protocols.**

| Specimen | Protocol | Lugol's concentration (%) | Vessel | Conditions | Weight (g) | Volume of solution (ml) | CT voxel height/ Width (mm) |
|---|---|---|---|---|---|---|---|
| MUTV2SFI | Skin and fur intact | 5 | Lightfast | | 1,571.6 | 14,000 | Day 0: 0.578 |
| | | | | | | | Day 28: 0.627 |
| | | | | | | | Day 56: 0.511 |
| MUTV2C | Control | 5 | Lightfast | | 1,231.2 | 6,500 | Day 0: 0.498 |
| | | | | | | | Day 28: 0.487 |
| | | | | | | | Day 56: 0.449 |
| MUTV2A | Agitation | 5 | Lightfast | Agitated 2 x daily for 1 min | 1,150.0 | 6,000 | Day 0: 0.569 |
| | | | | | | | Day 28: 0.516 |
| | | | | | | | Day 56: 0.492 |
| MUTV2DVR | Decreased volume ratio | 5 | Lightfast | | 1,227.3 | 3,100 | Day 0: 0.499 |
| | | | | | | | Day 28: 0.507 |
| | | | | | | | Day 56: 0.505 |
| MUTV2DC | Decreased concentration | 2.5 | Lightfast | | 1,360.9 | 7,000 | Day 0: 0.477 |
| | | | | | | | Day 28: 0.385 |
| | | | | | | | Day 56: 0.491 |
| MUTV2IC | Increased concentration | 10 | Lightfast | | 1,359.0 | 7,000 | Day 0: 0.758 |
| | | | | | | | Day 28: 0.546 |
| | | | | | | | Day 56: 0.517 |

**Note:**
The protocol specifications for the MUTV2 possum specimens used in Stage Two of the experiments describing each specimens Lugol's iodine concentration, conditions, weight and volume of Lugol's iodine. The table also includes the voxel size of the CT at each imaging time point.

prior to the experiment (MUTV1C; 1,352 g total weight). Note, the specimens were not chemically treated prior to freezing. The specimen was thawed at room temperature for 72 h, then imaged to establish a baseline CT dataset for the specimen prior to tissue fixing and diceCT treatment (see below). At the completion of these scans, the specimen was skinned (excepting the ears, manus and pes), gutted to remove abdominopelvic organs, and placed into 10 L of 10% neutral buffered formalin for 7 days. Once the specimen was fixed, a Day 0 CT scan was taken. The specimen was then placed into a 20 L lightfast vessel at room temperature with 12 L of 5% Lugol's iodine and CT scanned every 7 days for 8 weeks.

The Lugol's iodine ($KI_3$) was mixed in 1 L batches as a cordial of 15% concentration using the following formula (*Nasrullah, Renfree & Evans, 2018*):

1 L of 15% Lugol's solution = 100 g potassium iodide powder (KI) + 50 g crushed iodine crystals ($I_2$) + 1,000 ml $H_2O$.

This cordial was stirred on a magnetic stirrer for 30 min until all the potassium iodide powder (KI) and crushed iodine crystals ($I_2$) dissolved. It was then diluted 1:2 with water to produce the final stain, a 5% solution. This process was repeated four times to achieve the required 12 L of 5% solution.

### Stage two

In our second stage experiments, we assessed the effect of previously published and/or common diceCT protocol variants across six *Trichosurus vulpecula* individuals (MUTV2) (Table 1). All specimens were eviscerated through a midsagittal abdominopelvic incision and submerged in a solution volume adjusted based on specimen body mass to maintain consistent ratios between the specimen and its stain solution (excepting MUTV2DVR; see below). We followed the same imaging, visualisation and quantification methods as described below; however, given the results of the Stage One experiments, only subjected the Stage Two specimens to CT scanning at the three time points (Days 0, 28 and 56) which were deemed essential to evaluate tissue radiodensity and volumetric changes.

The MUTV2SFI specimen was prepared and subjected to an identical diceCT protocol to the MUTV1C specimen in our Stage One experiment, except retaining all skin and fur to assess the extent to which this affected passive Lugol's iodine diffusion.

MUTV2C was established as a control specimen following the diceCT protocols to MUTV1C to establish the repeatability of the protocol in increasing tissue radiodensity across individuals.

The solution volumes were established based on mass. MUTV2SFI was heavier than MUTV1C so the solution volume was increased for this specimen to maintain the same ratio of Lugol's to specimen volume that MUTV1C was subjected to. The equation for this was 1.57 kg × 12 L/1.35 kg = 13.97 L. Of the remaining specimens (MUTV2A, MUTV2DVR, MUTV2DC and MUTV2IC), MUTV2DC was the heaviest at 1.36 kg so the equation was re-established using half the volume of MUTV2SFI (7 L). The decision to decrease the volume was based on the logistics of such high volumes and noticing that 7 L was more than sufficient in covering the specimen during the course of the experiment.

MUTV2A also mirrored MUTV1C and MUTV2C in the overall diceCT protocol but varied in having the container solution agitated twice daily every day for 56 days to assess whether reducing the settling of Lugol's iodine in solution impacted diffusion. MUTV2DVR was subjected to a reduced specimen: volume ratio by half. Finally, MUTV2DC and MUTV2IC assessed the effect of varying the concentration of Lugol's iodine solution on tissue diffusion through reduced (2.5%; MUTV2DC) and increased (10%; MUTV2IC) solution concentration.

### Imaging

All specimen imaging was undertaken at Monash Biomedical Imaging (Clayton, Australia). Both the baseline (pre-fixing) and all sequential CT scanning was undertaken with a Siemens Somatom go.UP using a standardised imaging protocol (130 kV, 350 mA, Sn filter, Pitch 0.35, rotation time of 1.5 s). Image stacks were reconstructed using the HR60 Kernel Reconstruction (Bone) setting with a slice depth and interval resulting in a 0.30 mm voxel depth. Individual specimen voxel lengths/heights varied slightly due to differences in the reconstruction bounding box between specimens and weekly scan passes (see Table 1).

Each specimen was CT scanned weekly for eight consecutive weeks, resulting in the generation of nine separate scans of the MUTVIC (*i.e.*, Days 0, 7, 14, 21, 28, 35, 42, 49, and

56) and three separate scans for each of the MUTV2 specimens (*i.e.*, Days 0, 28 and 56). Before each scan the specimen was removed from the Lugol's iodine solution and rinsed under tap water for ~30 s to remove excess Lugol's iodine from the abdominal cavity and from fur remnants to reduce artificially radiodense regions. The specimen was then manually dried with paper towel. We used both reference photos and foam props to reduce variation in body positions between scans.

### Visualisation and quantification of the CT data

In order to assess Lugol's iodine diffusion into specimen soft-tissues, we chose to evaluate changes in radiodensities occurring at the mid-thigh of the specimen. This location was selected for four primary reasons. First, the transverse (anteroposterior) cross-section of the thigh represents the largest thickness of skeletal muscle in the common brushtail possum, thereby representing a critical region for assessing 'complete' tissue diffusion. Second, the mid-thigh is largely homogenous skeletal muscle tissue with minimal tissue-type variation translating to potentially varying degrees of diffusion from the periphery to the femur. Third, the thigh represents a region not impacted by the decision to open the abdominopelvic cavity during evisceration (*e.g.*, is representative of diffusion in the animal irrespective of organ removal/tissue interruption due to our experiment design). And fourth, the mid-thigh is a region unlikely to be impacted by specimen movement or repositioning that allowed for straightforward, confident post-scan orthogonal reslicing of the CT dataset to standardize and facilitate direct comparisons across the sequential CT datasets.

This last point is particularly critical because sequential CT scanning and reconstruction with varying bounding boxes across 8 weeks will necessarily result in different specimen positions in the reconstructed datasets. To standardize our analysis of each CT dataset, we imported the reconstructed DICOM dataset into Materialise Mimics v.22 (Materialise Mimics, 20.0 ed.; Materialise NV, Leuven, Belgium). Using skeletal landmarks on the femur, we established an orthogonal axis along the long axis of the femur and extracted a resliced CT dataset for each of the individual scans undertaken across the 8-week experiment (see Fig. S1). This resampling effectively eliminated purely positional differences in sequential scans and allowed us to establish directly comparable datasets for our analysis.

One of the most important assessments was to establish the Lugol's iodine penetration over time using the changing radiodensities in Hounsfield Units (HU) over the course of the experiment. Utilizing the resampled dataset, we developed two quantitative assessments to track the progression of the Lugol's iodine penetration. First, we established a mid-thigh transverse cross-section from each week's dataset. To do this we used the greater trochanter of the hip and the lateral epicondyle of the femur as external landmarks on the specimen to measure the length of the femur to achieve a mid-thigh cross section. The resulting section was exported from Mimics to ImageJ (*Schneider, Rasband & Eliceir, 2012*), where we established an anteroposterior transect through the centre of the femur to plot tissue radiodensity levels (in HU) along this transect (in millimeters). This was to enable us to track the changes in HU over time (Fig. 1; Fig. S1).
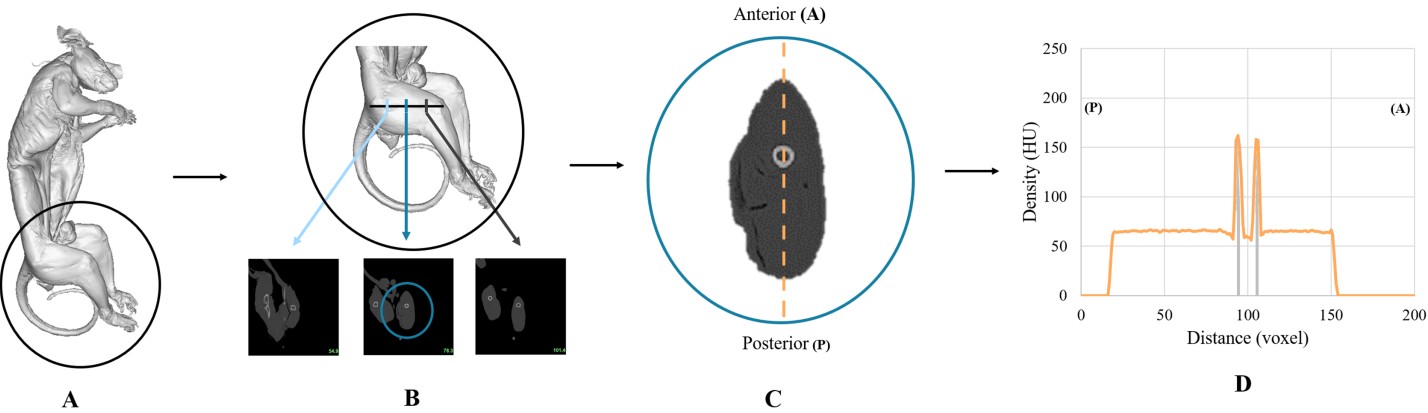

**Figure 1 A visual description of how the CT data was quantified.** (A) 3D render of the brushtail possum where (B) demonstrates the measurement between the greater trochanter of the hip and the lateral epicondyle of the femur to establish a proximal quarter, a midpoint, and a distal quarter. (C) Anteroposterior transect through the center of the femur that was used in (D) to track the radiodensity changes through the specimen over time. The two strong peaks shown either side of the 100-unit mark with the grey lines demonstrate the high radiodensity of the femoral diaphysis.

Second, we quantified soft-tissue volume reductions (shrinkage) in the thigh across the 8-week duration of the experiment. We used the maximum linear measurement of the MUV1 femur to divide the thigh into four segments and volumetrically reconstructed the slices between the midpoint of the thigh to the distal quarter of the thigh using segmentation tools in Mimics and volume calculation tools in Geomagic Studio 14 (3D Systems, Rock Hill, SC, USA) (Fig. S2).

All data including individual CT scans, midthigh cross sections used for assessing HU's and midthigh volumes used to assess volumetric changes are available on Morphosource under the project title *Experimental Assessment of Diffusible Iodine-Based Contrast-Enhanced Computed Tomography (DiceCT) Protocols Datasets* with specific DOIs found in Fig. S3.

## RESULTS

### Stage one

Progressive changes in tissue radiodensity in the MUTV1C specimen across the 8-week experiment are presented as a series of transverse CT images taken at the mid-thigh and the corresponding histograms of our Hounsfield Unit (HU) transects (Fig. 2). Qualitatively, there is a distinct increase in radiodensity and visual separation of individual muscle bellies from the baseline Day 0 (pre-treatment) CT scan to Day 35. The overall diffusion and increase in radiodensity progresses relatively evenly from the circumference of the specimen towards the center of tissue mass. There is no appreciable radiodensity changes across the Day 42 to Day 56 cross-sections, although muscle belly separation is apparent in these last three CT scans with the appearance of low-density regions within the posterior compartment (Fig. 2).

Our observations are supported by HU mapping across these transverse sections (Fig. 2). The Day 0 scan records only two high density peaks corresponding to the femoral cortical bone, but with sequential scans across Lugol's iodine exposure overall voxel

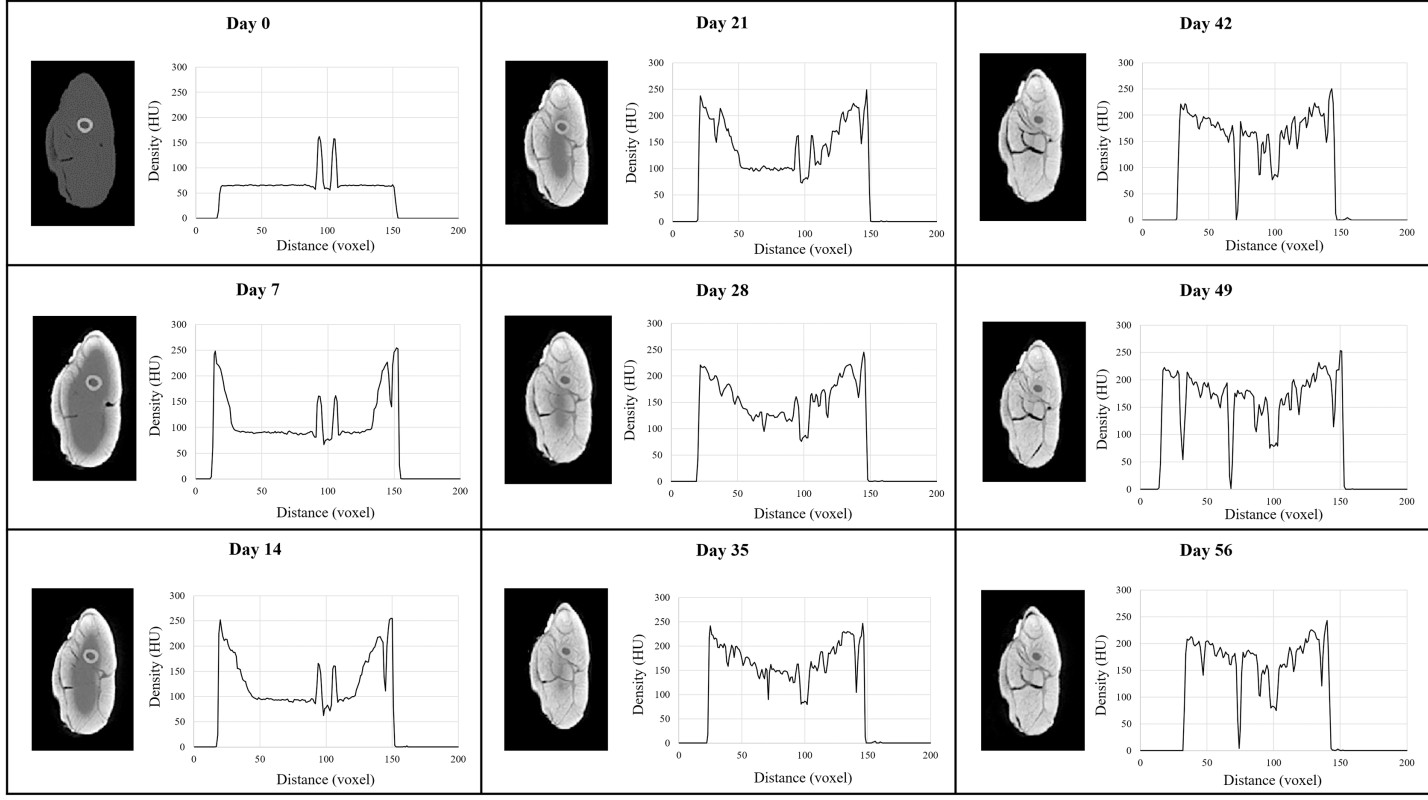

**Figure 2 MUTV1C comparison of weekly CT cross section and histogram data.** A side-by-side comparison of a single cross section taken through the midshaft of the thigh of MUTV1C every 7 days for 8 weeks and the matching histogram of density values through the thigh midshaft cross sections showing the Lugol's penetration across 8 weeks. The progression of the radiodensity as presented by the lightening of the structures can be seen providing more information on the muscular structures. From Day 42 to Day 56, there is visible stabilisation of the stain in the deepest tissues of the thigh.                                                           

radiodensity across the section increases from both the anterior and posterior margins of the thigh and reaches similar (or higher) HU values to cortical bone across the entire section by the Day 28–Day 35 CT scan. Minimal HU value changes occur beyond this scan point, with the exception of several depressed HU regions that reflect muscle belly separations. Simultaneously we note that despite using 200 voxel transects, the transition anteriorly and posteriorly from 0 HU values (*e.g.*, the air-tissue interface) appears to reduce. In sum, Fig. 3 plots the mean transverse cross-sectional HU value obtained in each CT scan to establish a trendline in radiodensity values across the duration of the MUTV1C experiment. The strongest, and expected, increase in mean HU values occurs between Day 0 and Day 7 with the first exposure to Lugol's iodine solution. The increases in mean HU values occurs steadily between Day 7 and Day 35; subsequent to the Day 35 scan there is a flattening of the uptake curve indicating saturation of specimen tissues and obtaining peak radiodensity after 5 weeks of treatment.

A plot of the specimen volumes extracted from the thigh of the MUTV1C specimen is provided in Fig. 3. Our Day 0, pre-treatment scan resulted in a calculated regional volume of 28.48 cm$^3$. By Day 7 there has been a 6.08% reduction in thigh-segment volume (to 26.74 cm$^3$). From Days 7 to 14, there was a further 2.12% reduction, from Days 14 to 21

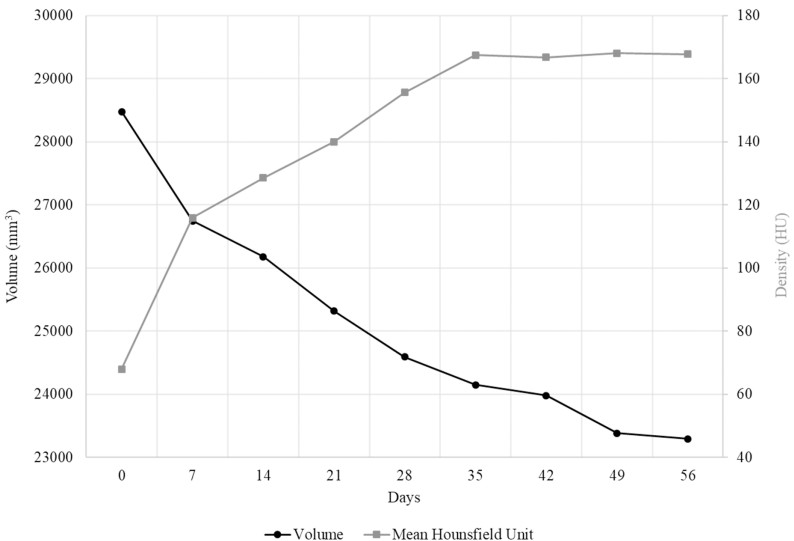

**Figure 3 The change in volume and radiodensity of MUTV1C over 56 days.** Combined specimen volumetric and density value changes in the thigh of the MUTV1C specimen. The black line with circle markers indicates the rate of volumetric change of a segment through the thigh of MUTV1C taken from the midshaft to the distal quarter. In comparison, the grey line with square markers shows the mean HU of a single cross section through the midshaft of the thigh over the eight successive weeks of Lugol's iodine penetration.

there was a further loss of 3.17% and between Days 21 and 28 another 2.98% reduction. This shows that in the first 4 weeks of penetration, there was an overall loss of 14.35% of soft-tissue volume within the segment analysed. Sequential weeks record a slow, but steady, decrease in thigh volume until reaching a measured volume of 23.29 cm$^3$ at Day 56 (Fig. 2). This reflects a total measured volumetric decrease of 18.21% to this soft-tissue segment over the 8-week experimental period.

## Stage two

As in Stage One, progressive changes in tissue radiodensity in the MUTV2 specimens across the 8-week experiment are presented as a series of transverse CT images taken at the mid-thigh (Fig. 4) at the three CT scan time points (Days 0, 28, and 56) and the corresponding histograms of our Hounsfield unit (HU) transects (Fig. 5).

## MUTV2SFI–skin and fur intact

Specimen MUTV2SFI retained the skin and fur, but was otherwise subjected to identical protocol conditions as MUTV1C within our Stage One experiment. The Day 28 transverse section demonstrates significant changes in radiodensity (Fig. 4A) and minimal overall changes in visible diffusion of iodine between Day 28 and Day 56 beyond some suggestions of low-density regions appearing between adjacent muscle bellies. Our HU transects (Fig. 5A) support these observations, with the Day 56 peak HU values above those of the femoral cortical bone. Although there are some differences in the HU value transects between the Day 28 and 56 scans, there is not the same development of low-density regions within the MUTV2SFI specimen as in MUTV1C.

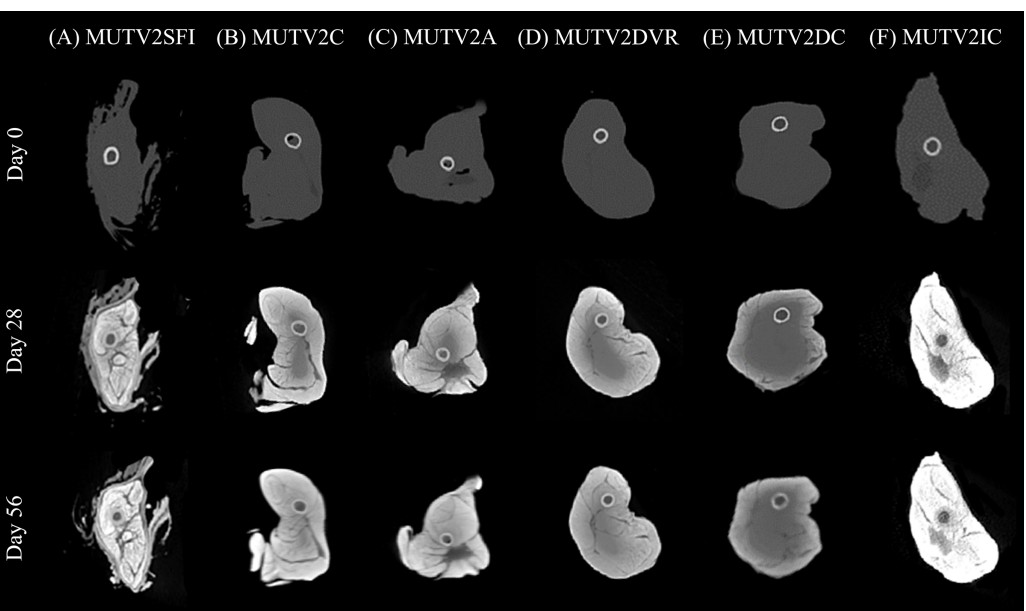

**Figure 4** **Lugol's iodine stain comparison of the stage 2 specimens CT scan cross sections.** A side-by-side comparison of cross sections through the midshaft of the thigh of the MUTV2 possum specimens from Days 0, 28 and 56 to visualise the Lugol's Iodine penetration. (A) MUTV2SFI, skin and fur; (B) MUTV2C, control 5% Lugol's concentration; (C) MUTV2A 5% Lugol's concentration agitated daily; (D) MUTV2DVR 5% Lugol's concentration with decreased specimen: solution volume ratio; (E) MUTV2DC, decreased (2.5%) Lugol's concentration; (F) MUTV2IC, increased (10%) Lugol's concentration.

## MUTV2C-control

Specimen MUTV2C represented the control specimen for our Stage Two experiment, reflecting application of the same protocol experimentally assessed for the MUTV1C specimen. The Day 28 transverse section demonstrates a significant increase in tissue radiodensity, but a central core of tissue that has not been penetrated through diffusion (Fig. 4B). By Day 56 all soft tissues had reached similar radiodensity with no obvious evidence of low-density gaps between muscle bellies (Fig. 4B). The HU histograms (Fig. 5B) broadly support these observations, though a distinct decrease in density values at the ~30 voxel point on the transect suggests some muscle belly separation.

## MUTV2A–agitation

Specimen MUTV2A received twice-daily agitation of the Lugol's iodine solution for the duration of the 8-week experiment. As in MUTV2C, the Day 28 transverse section indicates a low-density region central within the specimen that reflects a lack of solution diffusion that is only apparent in the final Day 56 scan (Fig. 4C). The HU transect (Fig. 5C) demonstrates minimal HU value changes or development of low-density regions in the thigh between the Day 28 and Day 56 scans. We note that specimen MUTV2A has an unusual, low-density region that is faintly visible in the Day 0 pre-treatment scan and persists across the experiment duration. While not ultimately increasing in radiodensity

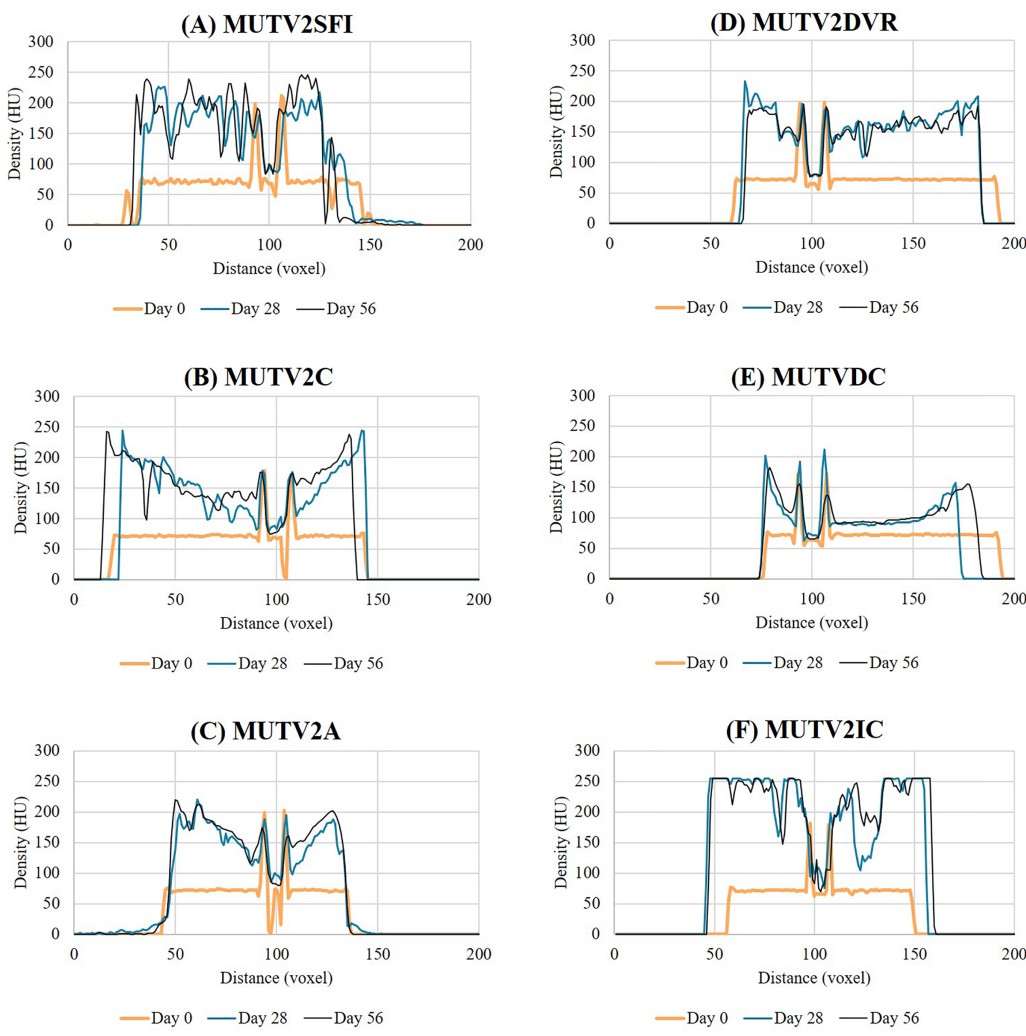

**Figure 5 Histogram data showing the Lugol's iodine penetration through the thigh of the stage 2 specimens.** Histograms through the thigh midshaft cross section of the MUTV2 specimens tracking the Lugol's penetration at Days 0, 28 and 56. (A) MUTV2SFI, skin and fur; (B) MUTV2C, control 5% Lugol's concentration; (C) MUTV2A 5% Lugol's concentration agitated daily; (D) MUTV2DVR 5% Lugol's concentration with decreased specimen:solution volume ratio; (E) MUTV2DC, decreased (2.5%) Lugol's concentration; (F) MUTV2IC, increased (10%) Lugol's concentration.

across the experiment, this low-density zone did not appear to impede Lugol's iodine diffusion around the rest of the thigh region.

## MUTV2DVR–decreased volume ratio

Specimen MUTV2DVR was exposed to a reduced ratio of Lugol's iodine volume relative to the prepared mass of the specimen. Evaluation of the resulting transverse sections and HU transects demonstrates a reduced increase in soft tissue radiodensity (Figs. 4D and 5D). While ultimately providing enhanced separation of individual muscle bellies, there is not a notable change in HU values across the transects taken at Day 28 and 56; and there is a distinct low-density zone persists in the posterior compartment muscle mass through the

Day 56 scan. Simultaneously there is no visible or detection of dips in HU values across the transect suggesting muscle belly separation.

### MUTV2DC–decreased concentration

Specimen MUTV2DC was exposed to a reduced (2.5%) concentration of Lugol's iodine solution. Both the transverse sections and HU transects document minimal increases in soft-tissue radiodensity–overall and between the Day 28 and Day 56 scans (Figs. 4E and 5E). The overall diffusion of Lugol's iodine appears to be restricted to the superficial ~25 voxel zone (or 10 mm) at the periphery of the thigh. The minimal increase in tissue radiodensity prohibited any ready evaluation of muscle belly separation or changes across the 8-week experiment duration.

### MUTV2IC–increased concentration

Specimen MUTV2IC was exposed to an elevated (10%) concentration of Lugol's iodine solution. Both the transverse sections and HU transects document that by Day 28 the thigh soft tissues have well-exceeded the femoral cortical bone radiodensity and high-density refraction artefacts are present in the CT data (Figs. 4F and 5F). There are no apparent HU value changes between the Day 28 and 56 scans. The high HU values and saturation of soft-tissues provided minimal contrast between muscle bellies and made interpreting any muscle belly separations difficult and potentially eliminated by diffusion-induced high-density artifacts.

### Volumetric comparisons of MUTV2

Soft-tissue volumetric changes across the MUTV2 experimental specimens were quantified, as in MUTV1C above, by segmenting a section of the thigh from the midpoint to the distal quarter from the Day 0 and Day 56 CT scans and calculating the segment volume. The volumetric changes in cubic millimeters were mapped and MUTV2DVR (decreased solution to specimen ratio) showed the least shrinkage with an 8.4% decrease in volume overall and MUTV2IC (increased Lugol's iodine concentration to 10%) showing the greatest shrinkage overall with a 35.5% decrease in volume. At Day 28 MUTV2DVR had decreased by 4.8% with the remaining 3.8% occurring from Days 28–56. MUTV2IC showed a similar trend of experiencing the greatest shrinkage in the first 28 days with a decrease of 32.1% and the remaining 4.9% in the remaining time to Day 56 (Fig. 6).

MUTV2A (twice daily agitation) and MUTV2DC (decreased Lugol's concentration to 2.5%) followed this trend also by showing larger decreases of 19.2% and 15.2% at Day 28 and a further 3.9% and 0.7% at Day 56 with overall volume losses of 22.3% and 14.6%

MUTV2C (control) and MUTV2SFI (skin and fur) bucked the trend showing a 1.3% and a 1.6% decrease in volume in the first 28 days and 11.4% in the last 28 days totaling an overall loss of 12.5% and 13% (Fig. 6).

## DISCUSSION

Our Stage One experiment established several important features of effective application of diceCT protocols in our mammal specimens. Our qualitative evaluation of CT scan transverse sections and quantitative evaluation of Hounsfield Unit changes across the

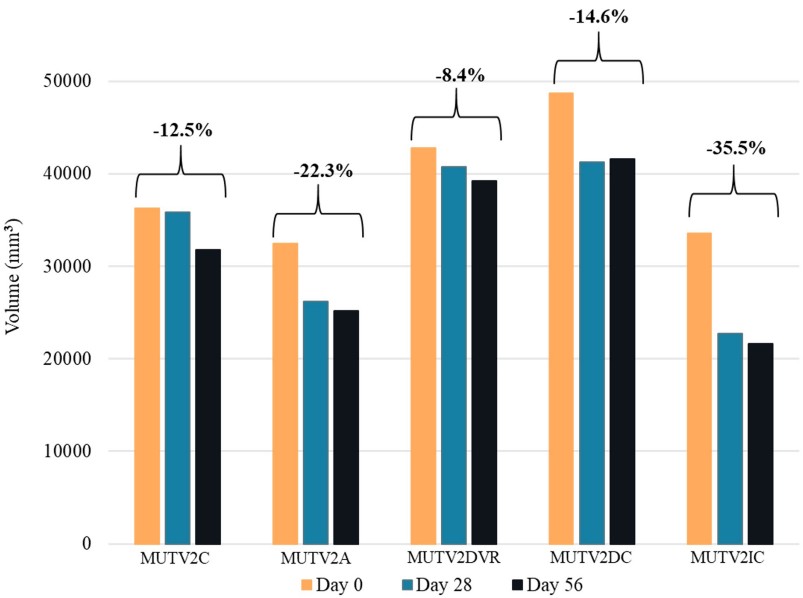

**Figure 6** **The change in volume of the MUTV2 specimens.** 3D volumes of distal thighs from protocol-testing possum specimens, showing reduction in volume (*i.e.*, specimen shrinkage) after 8 weeks. MUTV2SFI, skin and fur; MUTV2C, control 5% Lugol's concentration; MUTV2A 5% Lugol's concentration agitated daily; MUTV2DVR 5% Lugol's concentration with decreased specimen: solution volume ratio; MUTV2DC, decreased (2.5%) Lugol's concentration; MUTV2IC, increased (10%) Lugol's concentration.

duration of the experiment indicates that the useful transition in soft-tissue radiodensity occurred within the first 35 days (5 weeks) of exposure to the Lugol's iodine solution (Figs. 2 and 3). Beyond 35 days, not only are there no appreciable changes in HU values with continued exposure to the Lugol's iodine, but several visible and quantifiable changes occur in the specimen-represented by muscle belly separation and development of low-density regions within the thigh region (Fig. 2). There are several potential sources for these low-density areas, including artefacts introduced by manual handling of the specimen (*e.g.*, shifting tissues), gravity (*e.g.*, separation due to positioning in scanning), or our need to re-slice the datasets for standardising our imaging datasets. While all of those sources may contribute to muscle belly separation, we simultaneously were able to document soft tissue volumetric changes to the specimen across the duration of the experiment which may drive at least some of that observed separation (Fig. 3). The greatest volumetric change occurs in the first 7 days of exposure, occurring in tandem with the most significant change in soft tissue radiodensity (as assessed by mean HU value). Importantly, this relationship changes across staining duration, with volumetric changes persisting beyond the Day 35 scan (beyond which tissue radiodensity changes are minimal). This indicates that while some specimen shrinkage is an unavoidable feature of employing diceCT protocols, the impact of these volumetric changes can be mitigated through limiting the duration of staining.

Our Stage One experiment also established data on the rate of Lugol's iodine diffusion into soft tissues, though it has become apparent that specimen radiodensity increases do

not occur in an even, linear fashion (Fig. 3). From Days 0–7, there was an average volumetric decrease of 247.39 mm$^3$ per day, Days 7–14 was 80.8 mm$^3$ per day, Days 14–21 it changed by 123.03 mm$^3$ per day, Days 21–28 was 103.64 mm$^3$ per day, Days 28–35 was 63.7 mm$^3$ per day, Days 35–42 was 24.08 mm$^3$ per day, Days 42–49 was 85.21 mm$^3$ per day and Days 49–56 there was a volumetric decrease of 12.72 mm$^3$ per day. There are several possible explanations for this weekly variability in linear penetration, particularly if progressive uptake of iodine by soft-tissues is restricted or exploiting natural variation in tissues (*e.g.*, regionally variable/present/absent loose connective tissue, dense fascias, adipose, muscle). Although we cannot confidently establish a simple relationship between duration of Lugol's iodine exposure and linear soft tissue radiodensity increases, we do advocate for continued experimentation with larger specimens and regions with varying tissue compositions. This may ultimately offer a more reliable set of guidelines for establishing experimental designs to gauge the likely linear radiodensity increases over time in even larger specimens.

Our Stage Two experiments directly assessed a series of variables of interest when planning diceCT applications in larger mammal specimens, ranging from the potential impacts of retaining the skin and fur to exposure to varying concentrations and volumes of Lugol's iodine solution (Figs. 4 and 5). Our results from the MUTV2SFI specimen, where the specimen was submerged with skin and fur intact, does not suggest that either of these superficial tissues prohibit or significantly alter the diffusion of Lugol's iodine solution by deeper soft-tissues (in an eviscerated specimen). Our handling protocol of rinsing the specimen in water prior to CT scanning appears to have mitigated any concerns over retained Lugol's solution in fur creating refraction artifacts in the scans (*Gignac et al., 2016*). We documented effectively full radiodensity changes in the thigh musculature by our Day 28 CT scan, which mirrors the results obtained from both the skinned MUTV1C and MUTV2C specimens. Simultaneously, we note that (and in contrast to the MUTV1C specimen) there appeared to be minimal volumetric changes in the MUTV2SFI specimen–specifically between the Day 0 and Day 28 scans. Surprisingly MUTV2SFI, the specimen with the fur and skin still intact showed a reduced shrinkage rate than its comparative partner MUTV1C. MUTV1C had an overall shrinkage rate of 18.21% whereas MUTV2SFI has an overall shrinkage rate of only 14.6%. MUTV1C at Day 28 had decreased its volume by 14.35% whereas at the same time point, MUTV2SFI had only lost 1.6%. This demonstrates that retaining skin and fur on mammal specimens of this size provides the specimen with greater protection against volumetric shrinkage without compromising the application of the diceCT protocol for soft tissue radiodensity enhancement (*Gignac et al., 2016*).

This represents an important outcome and area for further experimentation, particularly when considering the application of diceCT protocols with sensitive mammal specimens (like those from museum wet collections). If retaining superficial tissues like skin and fur reduces volumetric changes when applying diceCT protocols, it provides a better argument for maintaining these tissues during diceCT applications and reinforces that unskinned, eviscerated specimens may be effectively used in diceCT research projects. What remains to be experimentally tested are the appropriate diceCT protocols for

enhancing the radiodensity of cranial, thoracic and abdominal viscera; particularly the latter region and the impact of maintaining the integrity of the anterior abdominal wall. We would also note that the process of evisceration and opening of the abdominal cavity in our study provided a potentially expedited diffusion path across an artificially exposed inguinal region into the thigh and/or vascular pathways between the abdomen and the thigh. This will likely be a factor in further experimentation, particularly when testing diffusion rates in modified specimens or isolated body regions.

Our experimental design included using the post-fixed specimen weight to determine the volume of Lugol's iodine solution and maintain consistent specimen:solution volume ratios across experimental animals. The retention of skin and fur on the MUTV2SFI specimen meant that, despite manually drying the specimen prior to weighing, the damp fur may have artificially inflated specimen weight (and therefore resulted in exposure to an increased solution volume). While a factor to consider, we do not believe this additional fluid mass would have resulted in such a significant effect as to distort the overall interpretation of skin and fur being neutral to Lugol's iodine diffusion described above.

MUTV2C was used as the control specimen for the protocol testing, and we observed diffusion of the Lugol's iodine through the specimen resulting in a strong contrast of all the musculoskeletal anatomy present in the thigh. MUTV2C experienced a 12.5% volume loss over the 8 weeks. When comparing the Day 56 cross sections with the other specimens, it stands out (with MUTV1C and 2C) as the most consistently diffused and easily to segment and gather anatomical information from.

Our results from the MUTV2A specimen, where twice daily solution agitation was performed, may support the suggestion (*Gignac et al., 2016*) that solution agitation would result in a more rapid diffusion of Lugol's iodine solution in larger mammal specimen soft tissues. When simply evaluating the resulting radiodensity changes in the cross sections on the Day 28 scan, there was little difference visually between MUTV2A and MUTV2C. But analysing the histogram and volumetric changes, it can be seen that MUTV2A did experience a greater uptake of the Lugol's in the first 28 days compared to the control specimen (MUTV2C) and the resulting volume loss was greater perhaps because of this sharper uptake. It is thought that the region in the thigh of MUTV2A that never became completely diffused with the Lugol's iodine was likely due to a tendon or neurovascular/fascia region carrying the sciatic nerve and associated vessels. The baseline scan also shows a physical gap in the specimen here too, so that would never be penetrated as there is no tissue there. That said, it is plausible that this degree of agitation was insufficient to expedite solution diffusion, particularly since prior diceCT applications (*Gignac et al., 2016*) in smaller specimens have had the benefit of using beakers and agitation plates. Given our focus on larger specimens, however, the practicalities of mechanical agitation do not at present seem warranted given the apparently minimal benefit.

Varying the relationship between the specimen and Lugol's iodine solution through reduced specimen:solution volume ratio, 1:2 (MUTV2DVR) demonstrated a similar visual outcome in soft tissue radiodensity increases as the control specimen (MUTV2C). It was also the specimen that experienced the least volumetric loss with a relatively even split from Days 0–28 and Days 28–56. With the reduced solution volume, such that the

specimen was only just immersed, there is a potential cost saving benefit particularly in research with even larger specimens or a series of specimens. This experimental protocol was also more effective at increasing soft-tissue radiodensity than an alternative cost-saving approach through reducing the concentration of Lugol's iodine (MUTV2DC), where a 2.5% solution was insufficient to reach diffusion through the thigh tissues at either the Day 28 or Day 56 scan points. While still presenting a volumetric decrease overall of 14.6%, the lack of full diffusion across the tissues led to retaining a relatively heterogenous radiodensity in cross section, rendering an unusable result.

Doubling the concentration of Lugol's iodine in solution to 10% (MUTV2IC) relative to the MUTV2C control demonstrated full (if not over-) saturation of soft-tissues by the Day 28 scan. By Day 28 a 32.1% volumetric reduction had already occurred (and by Day 56 had lost 35.5% of calculated specimen volume) resulting in again a homogenous radiodensity that made it hard to visualise the anatomy. While potentially affording a very short turnaround for enhancing the radiodensity of tissues, more frequent CT scan assessments would be needed to establish the optimal balance between concentration, duration, and volumetric changes to avoid compromising subsequent analysis.

Volumetric changes in the specimen are particularly critical to track and note, as shrinkage can affect calculations of *ex vivo* muscle form and volume, and associated downstream metrics (Figs. 4 and 6). Increasing the concentration levels to try and expedite the diceCT process does cause substantial volumetric changes. Although potentially acceptable in the context of some anatomical studies, we emphasise that it is critical to gauge or consider shrinkage introduced by diceCT when interpreting analysis results. While each experimental protocol will be shaped by individual research objectives, on balance our experimental results presented here would suggest that lower Lugol's iodine concentrations (2.5–5%) that are monitored weekly (ideally) *via* CT are more likely to achieve a combination of radiodensity enhancement and minimal volumetric changes to experimental specimens of this size.

## CONCLUSIONS

We present here a method for non-invasive quantification of larger-bodied mammal anatomy using experimentally based protocols. Lugol's concentration affected the soft-tissue radiodensity outcomes, with lower concentrations resulting in lower (likely inadequate) radiodensity changes and higher concentrations causing higher (likely unhelpful, overstained) radiodensity and volumetric changes, supporting a solution concentration of ~5% for specimens of this type over a 35-day treatment period. Quantification also showed that, on average, the greatest radiodensity changes (with the highest degree of tissue shrinkage and greatest solution depletion) occurred within the first 28 days of specimen exposure. This result supports the careful planning of diceCT treatment protocols, and suggests that establishing a baseline (pre-treatment) CT dataset of experimental animals can aid in establishing the degree to which volumetric changes have impacted resulting soft-tissue analyses. It also demonstrates the benefits of progressive CT-based monitoring of the specimen during the early exposure period to adjust for over or under-staining of the specimen. Based on our findings, theoretically the

best protocol for an unskinned, eviscerated mammal specimen of this type would be immersion in a 2.5–5% Lugol's iodine solution for no more than 5 weeks to optimise Lugol's iodine penetration *vs* shrinkage (with weekly monitoring). It is also recommended decreasing the specimen to stain volume ratio. Here we tested 1:2 this protocol saves on both space and material cost. In the future, further testing of diceCT protocol parameters in both larger mammal specimens and in specimens with varying tissue compositions will produce further baseline data for the technique which would help to optimise future project design. Using the quantification that was performed on the five different protocols tested here, it is hoped that more research groups will have the confidence to apply the diceCT technique to their research and have the ability to maximise data from their specimens with minimal expense and staining time.

## ACKNOWLEDGEMENTS

We acknowledge the facilities and scientific and technical assistance of Dr. Michael de Veer and Richard McIntyre at the National Imaging Facility (NIF), a National Collaborative Research Infrastructure Strategy (NCRIS) capability at Monash Biomedical Imaging (MBI), a Technology Research Platform at Monash University. We also gratefully acknowledge the assistance of Michelle Quayle, William Parker, Alexander McDonald, James Rule, and Douglass Rovinsky.

### Funding

This research was funded by internal operating funds provided by Justin W Adams in the Department of Anatomy and Developmental Biology at Monash University, Australia. The funders had no role in study design, data collection and analysis, decision to publish, or preparation of the manuscript.

### Grant Disclosures

The following grant information was disclosed by the authors:
Department of Anatomy and Developmental Biology at Monash University, Australia.

### Competing Interests

The authors declare that they have no competing interests

### Author Contributions

- Lucy F. Costello conceived and designed the experiments, performed the experiments, analyzed the data, prepared figures and/or tables, authored or reviewed drafts of the article, and approved the final draft.
- Hazel L. Richards conceived and designed the experiments, performed the experiments, prepared figures and/or tables, authored or reviewed drafts of the article, and approved the final draft.
- Alistair R. Evans conceived and designed the experiments, authored or reviewed drafts of the article, and approved the final draft.

- Justin W. Adams conceived and designed the experiments, analyzed the data, prepared figures and/or tables, authored or reviewed drafts of the article, and approved the final draft.

## Animal Ethics

The following information was supplied relating to ethical approvals (*i.e.*, approving body and any reference numbers):

This study makes use of brushtail possum (*Trichosurus vulpecula*) cadavers that were sourced through non-commercial and commercial sources as cadaveric specimens. No live animals were used in research, and therefore no Animal Ethics permissions were required. We note in the manuscript the sources of all specimens including relevant Victorian Department of Environment, Land, Water & Planning Permit number 10008717 that applies to the appropriate specimens.

## Data Availability

All data files including CT scans, Midthigh volumes and cross sections are available at Morphosource: https://www.morphosource.org/projects/000611683.

The 81 DOIs are in the Supplemental File.

## Supplemental Information

Supplemental information for this article can be found online at http://dx.doi.org/10.7717/peerj.17919#supplemental-information.

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
