# Peer review of "Experimental assessment of diffusible iodine-based contrast-enhanced computed tomography (diceCT) protocols"

_PeerJ, doi:10.7717/peerj.17919_

## Round 0.1 · original submission · Minor Revisions

The two reviewers have given speedy and constructive reviews, agreeing that the paper is a great contribution to the literature, which I concur with. They also both feel that the paper only needs some minor changes, although these have substance rather than merely presentational (but require no further analyses). I commend the reviewers for their very helpful reviews, and look forward to receiving the revised MS. Further review is unlikely to be needed if the revisions are attentive.

Reviewer 1 ·

Basic reporting

The article addresses a gap in current dice-CT literature testing a variety of protocols to stain large mammalian specimens in water-based solutions.
The text is clearly written, the figures and supplemental data are clear overall and the raw data is shared. The hypotheses and goals are clearly stated and the results address them.
While the background provided is sufficient, the authors seem to have missed a key paper that addresses iodine staining in large mammals, Lanzetti and Ekdale 2021 (https://onlinelibrary.wiley.com/doi/full/10.1111/joa.13410). This paper, while focusing on ethanol-preserved specimens, presents protocol to stain large mammalian specimens including ones with fur. References to this work should be added to the Introduction and Discussion.
Additionally the figures captions should be revised as they lack elements. For example, in Figure 5 and 6 there is no description of the parts A to F.
In the methods section, Stage One currently is not very clear, as it references scanning time frames listed in the Imaging section. This can be adjusted by rephrasing a few sentences.

Experimental design

The experimental design is solid and well explained. The methods and replicates are sufficient to accomplish the stated goals. The question posed in important and fills a knowledge gap.

Validity of the findings

This work provides an example of a partially novel application of na established method, though testing it in a new area and filling a knowledge gap. The conclusions are supported by the results and they are clearly stated while also raising possible new avenues of investigation.
All data have been provided, including the CT images uploaded to MorphoSource which I commend.

Reviewer 2 ·

Basic reporting

Costello et al.

This paper presents a very well-done sensitivity analysis of diceCT protocols by varying iodine concentration, skinning/no skinning, and solution agitation to examine specimen shrinkage and iodine diffusion. I think that this paper will be very valuable to researchers employing diceCT in their studies. I have no major changes, but a few comments to further enhance the paper and also grammar corrections. I feel that this paper could be published after minor revisions and would be happy to review a revised version.

You reference throughout the paper that these possums are 1-4.5kg. However, it is a little disingenuous because the largest one that you scanned in the study is ~1.57kg. I would make this clearer throughout. Further, to emphasize the larger size of the specimens in your study, I think it would be good to add a new figure showing the range of sizes scanned by other studies and then where your study is on that range. Maybe a horizontally oriented bar graph with different studies on the y axis and mass ranges on the x?

Referring to skinning, I have two thoughts. First, you eviscerated the specimens, which may have increased perfusion. You mention that this was a part of the plan at the beginning, but I wonder if you might comment using other works on whether this increased perfusion. Even though the thigh is not open to the abdominal cavity, it may be easier for the I2KI to travel down through the inguinal region via the abdominal cavity than perfuse across the skin. Second, I wondered what effect you thought removing limbs (or decapitating heads) might have. Scanning full specimens versus parts of specimens is variable across studies and since you are focused on larger specimens, I wonder if most people reading your paper will be scanning legs (etc.) rather than entire specimens. I don’t think any further experimentation is necessary, but this commentary may make for a nice additional paragraph in the discussion.

I just wanted to check that these specimens were not kept in any preservative before freezing. That seems to be something that can impact specimen shrinkage. You state that they were purchased and kept at -20ºC so I think they were fresh, but adding a few words in would help clarify.

I also think that figures 3 and 4 could be combined as a double y-axis plot. That would show how radiodensity and shrinkage relate to one another nicer than having them as two separate plots.

Figure 5 needs more explanation in the caption for what (A), (B), etc are. It would also be nice to show this in the figure as well as the caption.

Line by line:

Line 49: ‘was the development of diffusible..’

Line 84: I think reword this. Mammalogists are interested in tons of sizes of mammals including rhinos and elephants. Maybe this is expanding the upper bound of previous diceCT studies, but having the figure I mentioned about would drive this home.

Line 114: Need a space ‘the ‘MUTV’’

Line 113: I like this numbering system. Really helps clarify what specimen you’re talking about to the reader. The proofs folks need to make sure the table appears in this area though.

Line 434: ‘showed a reduced shrinkage rate compared to..’ rather than a less shrinkage rate

Line 439: This makes sense to me because the muscle is held to the skin by fascias and doesn’t have the potential to shrivel up as easily. Nice to see this confirmed.

Line 440: Need a period at end of sentence

Experimental design

See attached PDF

Validity of the findings

See attached PDF

Additional comments

See attached PDF

Annotated reviews are not available for download in order to protect the identity of reviewers who chose to remain anonymous.

---

## Round 0.2 · accepted · Accept

I have checked the revised MS and the authors have nicely addressed all of the reviewers' comments, with good justifications where they did not fully follow recommendations. I commend the authors on a good job revising and recommend publication of the paper. Congratulations on this rigorous and broadly useful study!